# Outcome of BCG Vaccination in ADA-SCID Patients: A 12-Patient Series

**DOI:** 10.3390/biomedicines11071809

**Published:** 2023-06-24

**Authors:** Daniele Canarutto, Chiara Oltolini, Federica Barzaghi, Valeria Calbi, Maddalena Migliavacca, Francesca Tucci, Vera Gallo, Giulia Consiglieri, Francesca Ferrua, Salvatore Recupero, Maria Celia Cervi, Hamoud Al-Mousa, Anna Pituch-Noworolska, Chiara Tassan Din, Paolo Scarpellini, Paolo Silvani, Claudia Fossati, Miriam Casiraghi, Daniela Maria Cirillo, Antonella Castagna, Maria Ester Bernardo, Alessandro Aiuti, Maria Pia Cicalese

**Affiliations:** 1Faculty of Medicine and Surgery, Vita-Salute S. Raffaele University, 20132 Milan, Italy; 2San Raffaele Telethon Institute for Gene Therapy (SR-Tiget), IRCCS San Raffaele Scientific Institute, 20132 Milan, Italy; 3Pediatric Immunohematology Unit and BMT Program, IRCCS San Raffaele Scientific Institute, 20132 Milan, Italy; 4Clinic of Infectious Diseases, IRCCS San Raffaele Scientific Institute, 20132 Milan, Italy; 5Pediatric Infectious Diseases Division, Department of Pediatrics, Ribeirao Preto Medical School, University of Sao Paulo, Sao Paulo 05508-000, Brazil; 6Department of Pediatrics, King Faisal Specialist Hospital and Research Center, Riyadh 11564, Saudi Arabia; 7Department of Immunology, University Children’s Hospital, 30-663 Krakow, Poland; 8Department of Anesthesia and Critical Care, IRCCS San Raffaele Scientific Institute, 20132 Milan, Italy; 9Emerging Bacterial Pathogens Unit, Division of Immunology, Transplantation and Infectious Diseases, IRCCS San Raffaele Scientific Institute, 20132 Milan, Italy

**Keywords:** enzyme replacement therapy, isoniazid, rifampin, tuberculosis, newborn screening, gene therapy

## Abstract

Vaccination with Bacillus Calmette–Guérin (BCG) can be harmful to patients with combined primary immunodeficiencies. We report the outcome of BCG vaccination in a series of twelve patients affected by adenosine deaminase deficiency (ADA-SCID). BCG vaccination resulted in a very high incidence of complications due to uncontrolled replication of the mycobacterium. All patients who developed BCG-related disease were treated successfully and remained free from recurrence of disease. We recommend the prompt initiation of enzyme replacement therapy and secondary prophylaxis to reduce the risk of BCG-related complications in ADA-SCID patients.

## 1. Introduction

The worldwide burden of pediatric tuberculosis (TB) cannot be understated, even if it is sometimes overlooked [1,2]. Universal vaccination with the Bacillus Calmette–Guérin (BCG) vaccine is recommended by the World Health Organization (WHO) as an effective measure to reduce the burden of TB [3]. In fact, it is adopted by most countries in which the incidence of TB is higher [4,5]. Patients with severe combined immunodeficiencies (SCIDs), however, are a notable exception to this rule: they should not receive vaccination with BCG as their underlying immune defect impairs control of the infection [3]. When this happens, about one half of them will incur BCG-related disease, often in the disseminated form, with high mortality [6]. In fact, especially in the absence of suggestive family history or newborn screening, the diagnosis of SCID is often formulated after the BCG vaccine has already been administered.

A large patient survey has reported on the impact of BCG in 349 patients who were vaccinated before SCID diagnosis (Table 1). Half of the cohort developed BCG-related disease, more often disseminated (34%) rather than local (17%). The risk factors for the development of BCG disease were T cells counts <250/uL and BCG vaccination in the first month of life. The latter was also predictive of BCG-related mortality. While the large majority of symptomatic children were treated, more than half of the asymptomatic children did not receive anti-tuberculosis therapy at the point of diagnosis with SCID.

Immune reconstitution inflammatory syndrome (IRIS) occurred in 1/3 (n = 55) of the 190 vaccinated patients undergoing hematopoietic stem cell transplantation (HSCT), mainly in the first month post-HSCT. Before HSCT, 47 of these 55 patients had BGC disease (33 with disseminated disease, 14 with localized BCG-itis), while 8 were asymptomatic. Notably, vaccinated children who received anti-tuberculosis therapy while asymptomatic had a significantly lower incidence of BCG complications before HSCT, reduced incidence of IRIS post-HSCT, and lower mortality from BCG-related complications [6]. 

The management of SCID patients who have been vaccinated with BCG is challenging, and there is no consensus on the drug regimens and the duration of the treatment, neither for BCG disease nor for vaccinated children who are still asymptomatic. This is a clinical challenge especially in view of allogeneic HSCT [7] or, in some cases, autologous hematopoietic stem and progenitor cell (HSPC) gene therapy (GT) [8]. These are the current definitive therapeutic options to correct the immune defect; however, they can give rise to a recrudescence of the BCG-related disease due to conditioning, aplasia, immunosuppression, and possibly also IRIS [9]. Adenosine deaminase deficiency (ADA-SCID) is among the most common forms of SCID and is peculiar in that the immune defect can be partially rescued by enzyme replacement therapy, which is often a bridge to HSCT or GT [10]. 

## 2. Aim of this Study

Here, we retrospectively reviewed the clinical course and management of patients with ADA-SCID who had been vaccinated with BCG and referred to our center, focusing on secondary prophylaxis, treatment of BCG disease, and long-term reactivation of BCG in order to guide decision-making in this particular setting.

## 3. Materials and Methods

We retrospectively reviewed the charts and case report forms of all consecutive pediatric ADA-SCID patients who were referred to the Istituto di Ricovero e Cura a Carattere Scientifico (IRCCS) Ospedale San Raffaele (Milan, Italy) between 1st January 2000 and 31 December 2022. Written informed consent from the parents or legal guardians was collected according to the Italian law. Four patients had been enrolled in GT clinical trials [11,12]. All studies were approved by the OSR Ethical Committee and the Italian competent authorities. BCG-related disease was classified as local, regional, distant, or disseminated, according to the pediatric criteria formulated by Hesseling and colleagues [13]. Briefly, the classification can be summarized as follows: (1) local BCG disease: BCG injection site abscess ≥10 mm × 10 mm or severe BCG scar ulceration; (2) regional disease: involvement of any regional lymph nodes (enlargement, suppuration, fistula formation) or other regional lesions beyond the vaccination site (ipsilateral axillary, supraclavicular, cervical, and upper arm glands); (3) distant disease: involvement of any site beyond a local or regional ipsilateral process that is BCG confirmed from at least one distant site beyond the vaccination site (for instance: pulmonary secretions, urine, cerebrospinal fluid, osteitis, distant skin lesion); (4) disseminated disease: BCG confirmed from more than one remote site (as described for distant disease) and/or form at least one blood or bone marrow culture. Weight percentiles were calculated according to the WHO charts. Anti-tuberculosis therapy administered to vaccinated children while asymptomatic was defined as secondary prophylaxis. A one-sided Fischer’s exact test was performed with Prism 9.4.1 (Graphpad Software, Boston, MA, USA). A *p*-value < 0.05 was considered statistically significant.

This study was approved by the Ethics Committee of Ospedale San Raffaele on 19/07/2017 (code 01/2017), and all the patients signed an informed consent form before enrollment in this study.

## 4. Results

Among the 55 ADA-SCID patients referred to our center over the last 22 years, the BCG vaccination status was available for all but three (94%). Twelve patients (21%) had been vaccinated with BCG (female = 5, see Table 2). The median age at ADA-SCID diagnosis was 6.5 months (range 2–21 months), and pegylated adenosine deaminase (PEG-ADA) enzymatic replacement therapy (ERT) was started with a median delay of 1 month after the diagnosis (range 0–14 months) in all patients but one. The BCG vaccination was always administered before the underlying immune defect could be diagnosed. Seven of these patients were vaccinated within the first month of life and two within the first 6 months, while, for three of them, the exact date could not be determined. The BCG strains were not documented but could usually be inferred from the BCG World Atlas [4].

Four patients (#9–#12), all born after 2017, received prophylactic therapy after being diagnosed with ADA-SCID, either with isoniazid (I) and rifampin (R, n = 3) or I + R + ciprofloxacin (n = 1), for 5 to 10 months, which was successful in preventing the development of BCG-related disease. Of the remaining eight patients, only half remained free from BCG reactivation (#5–#8), while patients #1, #2, #3, and #4 developed BCG-related disease and are reported in detail hereafter. These four patients all had consanguineous parents, failure to thrive, and a history of severe infections, along with other comorbidities. The disease was local (n = 1), regional (n = 1), distant (n = 1), or disseminated (n = 1). The start of ERT was delayed by 6 or more months in three of these four patients due to unavailability of the drug. 

Patient #1 was diagnosed with ADA-SCID aged 6 months, following the second hospitalization for recurrent infections and failure to thrive. Due to the unavailability of PEG-ADA and of an HLA-matched donor, she underwent a haploidentical T-cell-depleted bone marrow transplant from her mother at the age of 9 months, without conditioning. While the initial manifestations of local BCG disease appeared around the age of 9 months, the diagnosis could be made only 3 months after transplant when *M. bovis* resistant to isoniazid, pyrazinamide (intrinsically resistant), and ethionamide could be isolated from the BCG inoculum site. While the resistance to isoniazid was later disputed, treatment was initially withheld to limit the systemic toxicity of an isoniazid-free regimen, hoping for spontaneous resolution of the local lesion. However, the subsequent development of a cutaneous abscess requiring surgical drainage led to reconsideration of this decision and administration of pharmacological treatment with ciprofloxacin, rifampin, and ethambutol for 6 months. Unfortunately, HSCT engraftment failure was documented 160 days after transplantation, but the patient was successfully treated with GT at the age of 22 months.

Patient #2 was conceived with assisted reproductive technology and was affected by gonadal dysgenesis. He was diagnosed with ADA-SCID at the age of 2 months after enduring recurrent infections. He then received an unconditioned, T-cell-depleted haploidentical HSCT from his mother at the age of 5 months that did not engraft. The child developed distant BCG disease with cutaneous and sub-cutaneous localizations on the left shoulder, axilla, chest wall, and neck, with pleural and lung involvement (Figure 1). Initially, he received an anti-mycobacterial treatment with isoniazid, rifampin, clarithromycin, and ciprofloxacin. After four months, due to poor control of the infection and failure to thrive, the patient was switched to isoniazid, rifampin, and ethambutol for six additional months (as well as amikacin for two weeks). Having been transferred to Italy from his home country, the patient was also placed on a short course of ERT to better fight the ongoing infection. Isoniazid alone was then continued for 4 additional months following the completion of the aforementioned regimen. Gene therapy, initially foreseen at age 16 months, was delayed due to staphylococcal contamination of the bone marrow harvest and was successfully administered only 6 months later, after a second low-dose busulfan conditioning regimen. No further recrudescence of BCG-related disease was observed, despite the first administration of busulfan being followed by the infusion of the unmanipulated bone marrow backup.

Patient #3 was diagnosed with ADA-SCID at the age of 4 months following recurrent airway infections. Regional BCG disease (ipsilateral axillary gland) was diagnosed at the age of 7 months, and the child received anti-mycobacterial treatment with isoniazid, rifampin, and ethambutol, and subsequently with isoniazid alone, for a total of 18 months, alongside ERT three months later. Gene therapy was administered at the age of 19 months, without recrudescence of BCG disease. 

The history of patient #4 has been previously described [14]. Briefly, she developed disseminated BCG disease with cutaneous bilateral abscesses on the thighs, i.e., at the sites of administration of PEG-ADA, as well as osteolytic bone lesions and a granuloma of the central nervous system. She was treated with an anti-mycobacterial treatment consisting of an induction phase with isoniazid, rifampin, ethambutol, and moxifloxacin for 12 months; then, 7 months after the beginning of induction phase, she underwent gene therapy. Unfortunately, the GT failed to engraft, and she was deemed eligible for allogeneic HSCT. Thereafter, the anti-mycobacterial therapy was simplified and transitioned to a maintenance phase with isoniazid and rifampin, and, 6 months later, the child underwent three alpha–beta and CD19-depleted haploidentical HSCTs while still receiving isoniazid and rifampin. The first two, from the father, incurred, respectively, primary and secondary graft failure with macrophage activation syndrome, requiring treatment with the human anti-interferon γ antibody emapalumab, and probable pulmonary invasive aspergillosis. In the absence of BCG disease recrudescence—despite treatment with emapalumab—the maintenance therapy was temporally withdrawn due to serious drug–drug interactions between rifampin and cyclosporine A, which had been introduced before the third HSCT from the mother, which eventually engrafted. After the engraftment, an anti-mycobacterial maintenance monotherapy with isoniazid was re-instituted for a further 2 months and finally discontinued to due hepatotoxicity and adequate duration in the absence of BCG disease recrudescence. 

In all cases, the BCG vaccination had been administered before the diagnosis of ADA-SCID, and BCG-related disease appeared before the start of ERT. The treatment schemes reflect local practice, but, while heterogeneous, they were nevertheless successful in eradicating the disease. Patients #1, #2, and #3 were successfully treated with GT and remained free from BCG-related disease. Patient #4 was eventually rescued and remains alive and free from BCG disease recrudescence to date. Among the remaining patients (#5, #6, and #7) who were not diagnosed with BCG-related disease before GT, despite the absence of secondary prophylaxis, #6 and #7 were peculiar in that they were among the few with a normal weight at referral and were diagnosed later than the others, likely indicating a relatively milder disease phenotype. Instead, patient #8 died of EBV-related lymphoma at the age of 42 months, before she could be treated with GT, and we were unable to confirm or exclude BCG-related disease in her case. 

## 5. Discussion

As a referral center for the treatment of ADA-SCID with GT, we collected data on 12 patients vaccinated with BCG, coming from TB-endemic areas, mostly South America and the Middle East, where early universal BCG vaccination is standard practice. The four patients (33% of the cohort) who incurred BCG-related disease had failure to thrive and were not on ERT at the time of symptom onset, reflecting the limited availability of ERT outside selected high-income countries. Indeed, this indicates that even partial immunological reconstitution attributable to ERT is protective (*p* < 0.05) and should be started as soon as possible in ADA-SCID patients.

The administration of anti-tubercular therapy to asymptomatic BCG-vaccinated SCID patients is a matter of debate. In our cohort, in the absence of secondary prophylaxis, half of the ADA-SCID patients developed clinically relevant BCG-related disease (Figure 2), which required 6 to 18 months of combination therapy for eradication. The incidence is possibly underestimated due to the early death of patient #8 for unrelated causes. Conversely, while not statistically significant due to the small sample size (*p* = 0.14), secondary prophylaxis, together with ERT, was effective in preventing BCG disease in all patients. We, therefore, share the recommendation of a prompt prescription of secondary prophylaxis for all vaccinated children with ADA-SCID while asymptomatic with a combination anti-mycobacterial therapy [7] to prevent disease dissemination. Isoniazid plus rifampin for at least 3 months is a reasonable option, and the continuation of secondary prophylaxis for up to 6 months should be considered in severely immunocompromised patients (e.g., those with less than 0.250 × 10^9^ T-cells/L at diagnosis, who are at higher risk, as reported [6]). When locally available, rifapentine plus isoniazid allows for a shorter, 1-month long treatment in older patients, and administration to children under two years of age may become an option in the future [15,16].

While HSCT risks exacerbate BCG disease in SCID patients, we did not clearly observe BCG flares or IRIS during or after GT/HSCT, even in patients who did not receive anti-mycobacterial drugs prior to GT/HSCT, with two possible exceptions. In patient #1, the impact of HSCT was hard to account for given the absence of conditioning and engraftment. Meanwhile, in patient #4, affected by disseminated BCG infection, we observed a failure of GT, followed by the rejection of two haploidentical HSCTs and macrophage activation syndrome, which could possibly have been triggered by BCG itself [14,17]. The lower incidence of reactivation observed in our cohort as compared to >90% observed in a recent large case series of SCID patients treated with HSCT [7] may be due to a combination of factors. First, ERT can enhance immune control and elimination of BCG. Second, the conditioning regimen of GT is milder than that of HSCT, with consequent less profound immunosuppression, infectious risk, and toxicity. Third, the autologous source of HSPCs in GT abolishes the risk of graft-versus-host disease and, thus, the requirement for immune suppressive drugs, associated per se to an additional risk of infectious reactivation [7].

In summary, the World Health Organization strongly recommends universal BCG vaccination “as soon as possible” to reduce the burden [3] and incidence of childhood TB [18]. Moreover, BCG vaccination may also reduce non-tubercular infections related to early infant mortality [19]. Indeed, BCG is administered with various policies and strains in most countries of the world, usually with a single dose shortly after birth [4,5]. However, BCG vaccination can often harm children with SCID and should be delayed or avoided in newborns with a family history of known or potential immunodeficiency or severe adverse reactions to BCG. Unfortunately, SCIDs are usually diagnosed after BCG vaccination has already been administered, and patients are at risk of developing BCG-related disease with a possible unfavorable outcome. While, in theory, the vaccination of SCID patients could be prevented by implementing newborn screening for SCIDs [20] and delaying BCG vaccination until the results are available, in practice, such delays can be expected to be counterproductive due to missed appointments and reduced coverage. Instead, prompt administration of secondary prophylaxis and treatment of SCID may be an alternative viable strategy, especially in countries where newborn screening is not implemented. When considering definitive treatment options, the milder acute toxicity profile of GT may also reduce the risk of BCG-related disease as compared to HSCT.

## Figures and Tables

**Figure 1 biomedicines-11-01809-f001:**
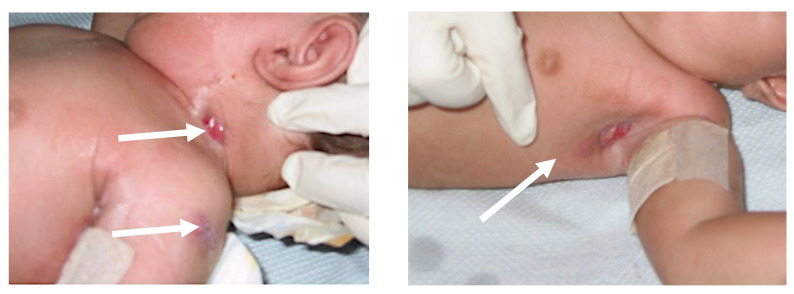
Distant BCG-related disease with visible lesions at the neck, left shoulder, and left axilla (white arrows) in patient #2.

**Figure 2 biomedicines-11-01809-f002:**
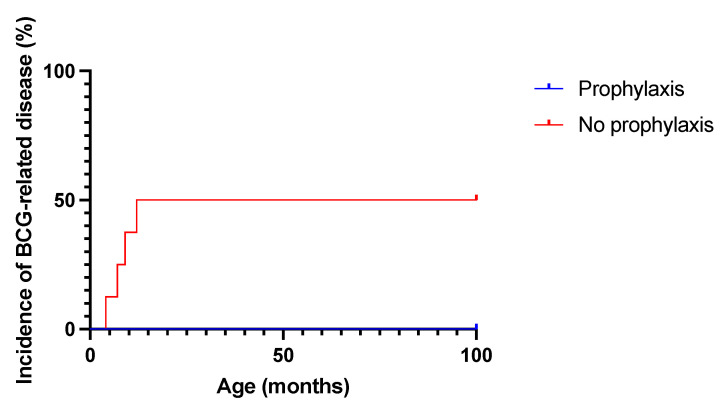
Incidence of BCG-related disease in patients receiving secondary TB prophylaxis or not (n = 8 in no prophylaxis group and n = 4 in secondary prophylaxis group).

**Table 1 biomedicines-11-01809-t001:** Disease course and therapy of patients affected by severe combined immunodeficiencies and vaccinated with BCG. Adapted from [6].

Cohort	n (%)	Anti-Mycobacterial Therapy
Vaccinated patients	349	238/349 (68%)
BCG-related disease	177/349 (51%)	160/177 (90%)
Disseminated	118/177 (34%)	107/118 (90%)
Localized	59/177 (17%)	53/59 (89%)
No BCG-related disease	172/349 (49%)	78/172 (45%)
**SCID treatment and IRIS**		
HSCT/GT/Thymus transplantation/ERT (% of total patients)	184/3/1/2 (54%)	
IRIS	55/190 (29%)	
Previously symptomatic for BCG-related disease	47/55 (85%)	

**Table 2 biomedicines-11-01809-t002:** Characteristics of the ADA-SCID patient population referred to IRCCS Ospedale San Raffaele and vaccinated with BCG. Strains in use according to bcgatlas.org: Tokyo 172-1 (Saudi Arabia), Danish 133 (Saudi Arabia), Moreau Rio de Janeiro (Brazil), Russia Moscow (Turkey), Pasteur 1173 P2 strain (Argentina and Saudi Arabia), Moreau (Poland), unknown for Venezuela [4]. ERT: enzyme replacement therapy. SDS: standard deviation score.

Patient #	Sex	Country of Birth	Lymphocyte Count before ERT (×10^9^/L)	Age at ADA-SCID Diagnosis/Age at Start of ERT (months)	Previous HSCT	Weight Centile at Referral	Comorbidities	Age at BCG Vaccination	Age at BCG Disease Onset (months)	Disease Classification	Treatment or (Secondary Prophylaxis)/Duration
1	F	Saudi Arabia	0.1	6/20	Yes	<5°	Lung interstitial disease, Arnold–Chiari malformation	Birth	9	Local (inoculum site)	Rifampicin, ethambutol, ciprofloxacin > ofloxacin/6 months
2	M	Saudi Arabia	0.14	2/8	Yes	<3°	Disorder of sexual development, bilateral peripheral hearing loss, psycho-motor retardation, hypothyroidism, pes pronatus	Unknown	4	Distant (skin, lymph nodes, lung)	Isoniazid, rifampicin, ethambutol, amikacin/14 months
3	M	Brazil	<0.25	4/10	No	3°	Pulmonary stenosis, hypothyroidism, arterial hypertension	20 days	7	Regional (lymph nodes)	Isoniazid, rifampicin, ethambutol/18 months
4	F	Turkey	0.039 T-cells	12/13	No	<3°	Anemia	Birth	<12	Disseminated (skin abscesses, central nervous system)	Isoniazid, rifampicin, ethambutol, moxifloxacin/12 month + isoniazid, rifampicin/6 months [14]
5	M	Venezuela	NA	4/NA	Yes	3–5°	Phimosis, feeding disorder, hemangioma of the liver	5 days	NA	Asymptomatic	
6	M	Algeria	NA	21/22	No	48°	Hypothyroidism, increased IgE	28 days	NA	Asymptomatic	
7	M	Turkey	0.1	18/18	No	48°		2 months	NA	Asymptomatic	
8	F	Turkey	0.14	2/2	No	<−3 SDS	Ileostomy, lymphoma	Unknown	NA	Asymptomatic	
9	F	Argentina	0.11–0.47	7/8	No	<−3 SDS	Psychomotor development delay, congenital complex heart malformation	15 days	NA	Asymptomatic	(Isoniazid, rifampin)/5 months
10	F	Argentina	0.13	7/8	No	<−3 SDS	Psychomotor development delay	15 days	NA	Asymptomatic	(Isoniazid, rifampin, ciprofloxacin)/5 months
11	M	Poland	0.36	7/10	No	62°		Birth	NA	Asymptomatic	(Isoniazid, rifampin)/6 months
12	M	Turkey	0.10	2/3	No	17°		Unknown	NA	Asymptomatic	(Isoniazid, rifampin)/10 months

## Data Availability

Not applicable.

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
