# Peer review of "Outcome of BCG Vaccination in ADA-SCID Patients: A 12-Patient Series"

_biomedicines, 2023, doi:10.3390/biomedicines11071809_

Round 1
Reviewer 1 Report
The manuscript represents an interesting approach related to primary immune deficiency and BCG vaccination. The use of ADA therapy is also interesting. The major problem is that the manuscript is difficult to follow. In the introduction, the second paragraph is confusing; it will help either to do a table illustrating the different responses or change the structure of the paragraph. The results are concentrated mainly on the first 4 patients. Why is there no much information on the rest? Patients 6 to 11 are quite older, and there is no information on several individuals but it is assumed that they are asymptomatic? It is unclear. The discussion also requires rewriting it is difficult to follow.
The manuscript requires editing. Some parts of the text are very difficult to follow.
Author Response
We thank reviewer #1 for his/her comments. We have rewritten the second paragraph and the discussion to improve readability as suggested. Given that the focus of the article is BCG-disease in ADA-SCID, we have reported in detail only the clinical picture of patients that were affected by BCG-related disease. the reviewer correctly noted that patient #6 and #7 were diagnosed later in life (and had higher BMI); we speculate that this indicates a milder disease phenotype as these patients did not develop BCG-related disease despite the absence of secondary prophylaxis.
Reviewer 2 Report
Given that BCG is administered in the first days of life, when the presence of congenital immunodeficiency, as a rule, is still unknown, it is the BCG complication that can become the first clinical manifestation of primary immunodeficiency. BCG vaccination should be postponed in newborns with a family history of severe complications to BCG vaccination, primary immunodeficiency, death of children at an early age from infections until the immune status is clarified. In the future, the introduction of neonatal screening for severe combined immunodeficiency may be promising in terms of avoiding serious complications of the BCG vaccine in this category of patients. In the article under review the authors analyze the outcome of BCG vaccination in kids affected by adenosine deaminase deficiency (ADA-SCID). The authors' suggestion of prompt administration of secondary prophylaxis and treatment of SCID seems to be a solid alternative strategy to neonatal screening in low-income countries.
Author Response
We thank the author for the positive comments on the manuscript
Reviewer 3 Report
This is an interesting, well organized article Canarutto et al. I would like to address a small number of suggestions to you that may improve the manuscript.
General recommendations
Please check all the text for spelling mistakes.
Page 2,
Please describe the abbreviation “HSCT” Line 57 and delete at the line 67.
Line 73-74
The sentence "Adenosine deaminase deficiency (ADA-SCID) is among the most common forms of 73 SCID, and peculiar in that the immune defect can be partially rescued by enzyme replacement therapy, which is often a bridge to HSCT or GT" may remove at the previous paragraph and aim of this study may be mentioned as separate paragraph.
Author Response
We thank reviewer #3 for his/her comments. We have checked the text for spelling mistakes, fixed the HSCT abbreviation as suggested, and added a new paragraph entitled “aim of the study”, as suggested.
Round 2
Reviewer 1 Report
The article has not been improved as requested. There were only minor changes in the introduction and discussion.
The English language was improved partially
Author Response
We have further polished introduction and discussion as requested by reviewer #1, including a table in the introduction as suggested.
